# Digital technology to facilitate Proactive Assessment of Obesity Risk during Infancy (ProAsk): a feasibility study

Sarah A Redsell,[1] Jennie Rose,[1] Stephen Weng,[2] Joanne Ablewhite,[3]
Judy Anne Swift,[4] Aloysius Niroshan Siriwardena,[5] Dilip Nathan,[6]
Heather J Wharrad,[7] Pippa Atkinson,[8] Vicki Watson,[9] Fiona McMaster,[10]
Rajalakshmi Lakshman,[11] Cris Glazebrook[12]

## ABSTRACT

**Objective** To assess the feasibility and acceptability of using digital technology for Proactive Assessment of Obesity Risk during Infancy (ProAsk) with the UK health visitors (HVs) and parents.

**Design** Multicentre, pre- and post-intervention feasibility study with process evaluation.

**Setting** Rural and urban deprived settings, UK community care.

**Participants** 66 parents of infants and 22 HVs.

**Intervention** ProAsk was delivered on a tablet device. It comprises a validated risk prediction tool to quantify overweight risk status and a therapeutic wheel detailing motivational strategies for preventive parental behaviour. Parents were encouraged to agree goals for behaviour change with HVs who received motivational interviewing training.

**Outcome measures** We assessed recruitment, response and attrition rates. Demographic details were collected, and overweight risk status. The proposed primary outcome measure was weight-for-age z-score. The proposed secondary outcomes were parenting self-efficacy, maternal feeding style, infant diet and exposure to physical activity/sedentary behaviour. Qualitative interviews ascertained the acceptability of study processes and intervention fidelity.

**Results** HVs screened 324/589 infants for inclusion in the study and 66/226 (29%) eligible infants were recruited. Assessment of overweight risk was completed on 53 infants and 40% of these were identified as above population risk. Weight-for-age z-score (SD) between the infants at population risk and those above population risk differed significantly at baseline (−0.67 SD vs 0.32 SD). HVs were able to collect data and calculate overweight risk for the infants. Protocol adherence and intervention fidelity was a challenge. HVs and parents found the information provided in the therapeutic wheel appropriate and acceptable.

**Conclusion** Study recruitment and protocol adherence were problematic. ProAsk was acceptable to most parents and HVs, but intervention fidelity was low. There was limited evidence to support the feasibility of implementing ProAsk without significant additional resources. A future study could evaluate ProAsk as a HV-supported, parent-led intervention.

For numbered affiliations see end of article.

**Correspondence to**
Professor Sarah A Redsell;
sarah.redsell@anglia.ac.uk

## Strengths and limitations of this study

► This study was the first to examine the feasibility of using the Infant Risk of Overweight Checklist prediction algorithm to differentiate between infants at population and above population risk of being overweight.

► Qualitative interviews providing both parent's and HVs' perspectives on the feasibility of conducting a future randomised controlled trial and implementing Proactive Assessment of Obesity Risk during Infancy within the proposed design were a study strength.

► The successful recruitment of some parents from socially deprived areas demonstrated that obesity prevention interventions can be implemented in hard-to-reach populations.

► The main challenges were misinterpretation of participant eligibility by health visitors resulting in less than adequate recruitment and varying levels of intervention fidelity.

**Trial registration number** NCT02314494 (Feasibility Study Results)

## BACKGROUND

Obesity is a global public health challenge that affects all ages. In 2015, over 42 million children under the age of 5 were overweight.[1] In the UK in 2015/2016, over a fifth (22.1%) of children aged 4–5 years were either overweight or obese,[2] with the highest rates in those living in socioeconomically deprived areas[2 3] or of Asian or Black ethnicity.[2] Although children are born with genetic predispositions related to weight and growth,[4] dietary behaviour is modulated by feeding experience and the family environment.[5] Interventions that address these practices have a role in childhood obesity prevention.[6]

Parents and health professionals have called for reliable and valid methods of identifying infants at risk of developing childhood

obesity.[7–9] This is possible because the risk factors for childhood overweight and obesity are identifiable antenatally and during infancy.[10 11] These include parental weight, smoking during pregnancy, birth weight and rapid weight gain; with breast feeding being moderately protective. Of these, the strongest risk factor for childhood overweight is rapid infant weight gain.[12–14] Between 25% and 33%[15 16] of infants gain weight more rapidly than desirable during the first 6 months of life[17–19] and this risk factor is potentially modifiable if identified during early life. The Infant Risk of Overweight Checklist (IROC),[12 20] developed using data from the Millennium Cohort Study,[21] and externally validated using the Avon Longitudinal Study of Parents and Children data,[22] operationalises an algorithm to predict an infant's future risk of being overweight. IROC[12 20] offers the opportunity to identify infants at greatest risk of overweight in clinical practice.

Interactive digital technology can support complex and sensitive discussions between health professionals and patients.[23–25] It is also suited to delivering personalised information about health status and risk.[26] Potential behaviour change can be raised through the neutrality of a technological device without increasing anxiety.[27] Proactive Assessment of Obesity Risk during Infancy (ProAsk) is a novel, interactive digital intervention designed to equip health visitors (HVs) with an individual infant's risk of future overweight, while also supporting discussions with parents. It incorporates the IROC[12 20] and a therapeutic wheel which has previously been found to be an acceptable format for patients with cancer.[28] The therapeutic wheel comprises an interactive graphic detailing strategies for preventive behaviour using supportive and strength-based prompts. The content of the therapeutic wheel is based on the behavioural strategies identified in a systematic review of interventions that reduce the risk of childhood obesity in early life.[6] ProAsk is designed to be delivered alongside motivational interviewing (MI), which has been successfully used to improve health behaviours including diet and physical activity for parent-child dyads.[29 30]

This is the first intervention designed to identify an infant's risk of overweight and to provide parents of infants at greatest risk with strategies for prevention. There is no evidence for the effectiveness of this approach to draw on to guide our study design, and so, in line with the Medical Research Council's framework for the of development and evaluation of complex interventions,[31] a feasibility study was required.

## METHOD
### Study aim and objectives
This study aimed to examine the feasibility and acceptability of undertaking a randomised controlled trial (RCT) of ProAsk with the UK HVs and parents. We used the ADePT (**A** process for **De**cision-making after **P**ilot and feasibility **T**rials) framework[32] to examine the

methodological feasibility, and to address the study's objectives which were:

i. to assess recruitment, response and attrition rates for parents/legal guardians/carers of infants;
ii. to determine the proportion of infants calculated as at risk using ProAsk at baseline to inform a sample size calculation for a future RCT;
iii. to evaluate the feasibility of HV delivery of the ProAsk intervention to eligible parents/legal guardians/carers, including an assessment of intervention fidelity and protocol adherence;
iv. to determine the acceptability and feasibility of the proposed primary and secondary outcomes measures.

### Design
Multicentre, pre- and post-intervention feasibility study with process evaluation.

### Participants and recruitment
ProAsk was delivered by HVs to parents in four study sites in two localities, in urban and rural deprived areas within the UK. One day's training in recruitment processes, ProAsk and MI[33] was delivered to members of the HV teams (n=47). Delivering the intervention required the skills of an HV but to ensure the team understood study processes we trained nursery nurses (n=12), an administrator (n=1), student nurses (n=3), managers (n=3), together with HVs (n=28). The MI training was delivered by a member of the motivational interviewing network of trainers (MINT) and comprised interactive and experiential activities[34] including agenda-mapping and reflective listening. HVs were also offered half-a-day top-up training in MI during the study period and six HVs from two sites attended. Recruitment commenced on 22 April 2015 in two sites and 12 June in the other sites, and ended on 30 November 2015.

The local NHS Trusts HV managers estimated that the numbers of births within the study areas over a 3-month period was 700. The National Institute for Health Research-Clinical Research Network estimated that HVs could recruit up to 30% of eligible parents to the study providing a pragmatic sample of approximately n=100 infants over a 3-month period.

Parents of infants aged 6–8 weeks were eligible to take part. Exclusion criteria were: (a) infant with known medical conditions requiring special diets, (b) mother with diagnosis of postnatal depression (PND) or score of moderate PND or above on HV applied screening tools (Edinburgh Postnatal Depression Scale >13)[35] (Patient Health Questionnaire-9 >10)[36] or anxiety (Generalised Anxiety Disorder-7 score >10),[37] (c) infant born before 32 weeks gestation, (d) infant birth weight below second centile and (e) parent with insufficient understanding of English to complete the questionnaires in the absence of face-to-face translation.

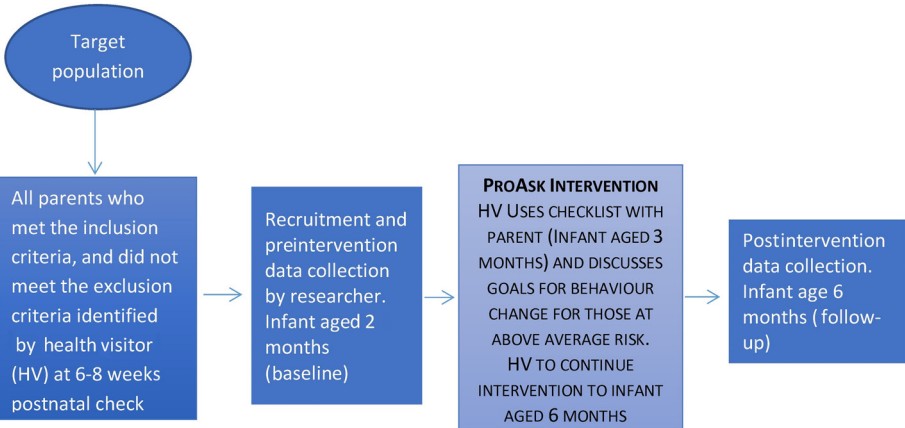

**Figure 1** Study regimen.

HVs were asked to identify all infants attending for their 6–8 weeks check over a 3-month period or until the sample size was met. They identified eligible parents at a routine infant check and approached them with information about the study. HVs recorded reasons for exclusion, response to approach, and reasons for refusal on a log sheet. Interested participants gave their permission to be contacted by the researchers by telephone to arrange a home visit, where informed written consent was obtained. Where parents did not respond to the initial phone call, the researchers made two further calls at different times of the day. At the end of the study, a purposively diverse sample of parent participants were invited to take part in qualitative interviews together with all of the HVs.

Figure 1 shows the per-protocol study regimen.

### Ethics and research governance permissions
Permission to conduct the study was provided by East of England (Essex) NHS Research Ethics Committee on 26 February 2015 (Reference number 15/EE/0011). Research governance permissions were provided by the two NHS Trusts covering the study localities.

### Intervention
HVs used a hand-held device (tablet) to deliver ProAsk to parents when their infants were aged 3 months. This involved entering the IROC[12] items (baby birth weight and length, current weight, maternal and paternal height and weight, maternal smoking status during pregnancy and breast feeding) into ProAsk, which then calculated the infant's risk status using the WHO growth charts.[38] This was displayed on the tablet screen as either "Your baby's risk of being above a healthy weight is the same as other babies" (population risk) or "Your baby's risk of being above a healthy weight is more than other babies" (above population risk). Responses were stored on the password-protected tablet. Two tablets were provided per site. Problems with internet access at two sites resulted in an amendment to the data extraction method and HVs were asked to screenshot the IROC result for transfer to the research team.

HVs were asked to offer parents who received the above population risk message an opportunity to explore the therapeutic wheel (figure 2). This interactive graphic promoted evidence-based behaviour change strategies[39] in four areas: active play; milk and solid foods; sleeping and soothing and infant feeding cues. It prompted HVs to use a motivational approach[33] to build parental self-efficacy for agreed behaviour goals, which were recorded on leaflets left in the home as cues to action for behavioural change.

### Measures and data collection
We recorded the number of participants identified by the NHS Child Health records and compared this with the numbers identified by the HVs. We also recorded the number of participants who were eligible, approached and recruited as well as the return of the follow-up measures.

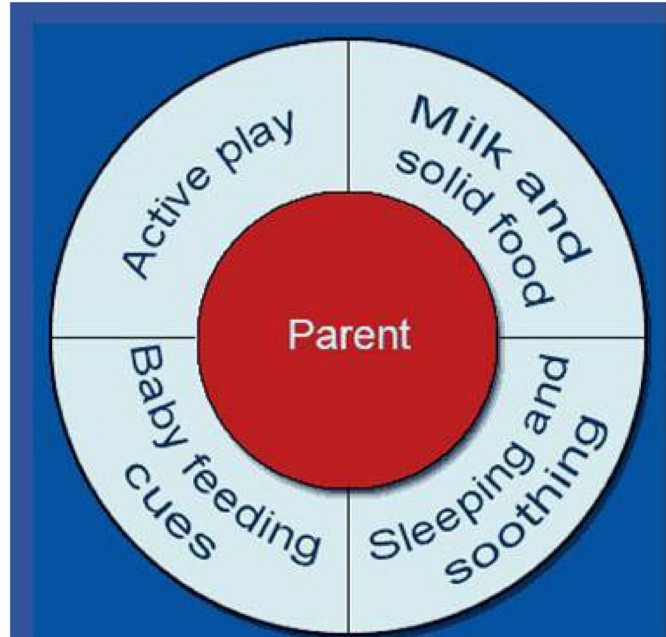

**Figure 2** Therapeutic wheel showing the options to support healthy weight.

A summary of the feasibility data collection measures can be found in online supplementary file 1. We collected data on the acceptability and feasibility of collecting data on the proposed primary and secondary outcome measures. The proposed primary outcome measure was weight-for-age z-score, using the WHO growth charts. The proposed secondary outcomes were parenting self-efficacy, maternal feeding style, infant diet and exposure to physical activity/sedentary behaviour.

Demographic details, ethnicity and information about family size were collected at baseline (infant aged 2 months) via a self-report questionnaire completed by parents. Details of the infants IROC score were recorded at 3 months. Infant anthropometric data, details of infant feeding (breast or formula milk or both) and validated measures of parenting self-efficacy[40 41] and maternal feeding style (Infant Feeding Questionnaire (IFQ))[42] were collected by self-report questionnaire at infant aged 2 months (baseline) and 6 months (follow-up). In addition, exposure to opportunities for physical activity and sedentary behaviour was recorded by parents at baseline and follow-up as time spent unrestricted on tummy, and restricted in a baby seat, car seat or pushchair.

Parents were interviewed about the acceptability of ProAsk, study processes, including recruitment and intervention fidelity. HVs were interviewed to explore their experiences of recruiting parents to the study and conducting the ProAsk Assessment. They were also asked about environmental factors such as the compatibility of ProAsk with existing workplace goals, organisational barriers and support for the intervention, and their views on the quality of training provided by the team. Interviews lasting up to 90 min were conducted face-to-face and over the telephone, and recorded using a digital Dictaphone.

### Data analysis

Recruitment, response and attrition rates, demographic details, weight-for-age z-score and overweight risk status (population risk vs above population risk at 10% risk threshold) were analysed using descriptive statistics via STATA V.13 MP4.

Audio data from qualitative interviews with parents and HVs were transcribed verbatim and transcripts were imported into Nvivo software for sorting, coding and categorising. Data relating to relevant methodological issues were subject to thematic content analysis using the method outlined by Boyatzis.[43] Verbatim quotes illustrate the themes.

### RESULTS

The results for each of the ADePT framework's[32] methodological issues are summarised in table 1, together with strategies for improving a future study design. The demographic data are detailed in table 2.

The results of the thematic content analysis of the parent (n=12) and HV (n=15) interviews are presented in table 3a and b.

### Sample size calculations

A total of 324 infants were screened by the HVs during a routine 6-week to 8-week check and consent was obtained from 66 parent-infant dyads (20%). An overweight risk assessment was completed for 56/66 infants and the data transferred to the research team for 53 of these. This showed that 40% infants were above population risk. Sufficient data were collected to inform a sample size calculation, but our findings suggest that more attention to study design is needed prior to future evaluation of ProAsk.

### Eligibility

The study flow chart presented in figure 3 details participant eligibility and the reasons for exclusion. The number of 6–8 weeks checks logged by the NHS Child Health Records during the extended recruitment phase was 589, which was fewer than the 700 estimated in 3 months by the NHS Trusts. HVs screened only 324 of these (45%) during the extended recruitment period (3–5.5 months for one locality and 7 months for the other).

In the HV interviews, language was identified as a major barrier to participant eligibility (n=9), particularly in one site (table 3a, n=7). HVs (n=8) were also concerned about referring parents with mental health, safeguarding or domestic violence issues.

#### HV N20

*It was the language barrier really. I'd say one hundred percent or ninety nine percent of my parents are non-English speaking so obviously without an interpreter.*

#### HV C37

*Because there were other issues around perhaps, safeguarding, in need, other agencies working with that family, and yet something else for them to have to deal with.*

### Recruitment

The recruitment target of N-100 infants in 3 months was not met. The most common reasons for parents declining were: parents not interested (n=28) and parents lacked time (n=21). The sample contained more than the expected number of mothers with degree-level education. The Income Deprivation Affecting Children Index (IDACI),[44] which measures area deprivation based on postcode, for participants with completed risk assessment showed that more (33%) of the participants recruited were from the two lower quintiles than from the two upper quintiles (25%) (table 2).

In total, 22/28 HVs who received training took part in the study (the remaining HVs were transferred, elsewhere, or on sickness or maternity leave). Most HVs interviewed took part at the request of their managers. Workload was identified as a barrier to parent recruitment by 5/15 HVs interviewed. Six reported being wary of raising the study with parents (table 3a).

Of the 12 parents interviewed, 11 found the study recruitment processes acceptable and 10 felt well-informed. Seven parents participated because of concerns

**Table 1** Summary of methodological issues

| Methodological issue | Findings | Evidence | Strategies for improvement |
|---|---|---|---|
| 1. Did the feasibility study allow a sample size calculation for the main trial? | Response rate less than expected but absolute numbers potentially allow for sample size calculation for main trial once a suitable design has been determined. | See figure 3. 66/324 (20%) eligible infants were recruited. 56/66 infants had an IROC assessment, but data from three assessments were missing due to data transfer errors, resulting in 53 complete assessments. 40% of infants assessed for overweight risk were above population risk. | Allow for longer recruitment time. Allow for more recruitment sites. |
| 2. What factors influenced eligibility and what proportion of those approached were eligible? | Fewer infants attending for 6–8 weeks check in the study areas. High number of screened infants were reported not to have met the inclusion/exclusion criteria (see figure 3). | HV managers at the NHS Trusts estimated there would be 700 infants born into the study areas within a 3-month period. NHS Child Health Records showed 589 estimated potential participants in the study period, of which 324 (45%) were identified by HVs. It is not known how many of the infants who were unrecorded by the HVs might have been eligible for the study. HVs recorded screening 324 (55%) parents. Of these, 226 (70%) were recorded as eligible for the study. | In the UK, it is not possible to employ dedicated researchers in one organisation to directly recruit participants to studies being undertaken in another organisation. Therefore, participant identification has to be undertaken via a gatekeeper employed by the organisation where the research is taking place. However, once participants are identified dedicated researchers can recruit directly (as was undertaken in this study). Investment in additional training in participant identification is required for HVs. This would require investment from the HVs employing Trusts in terms of time for training activities and continuing professional development. Researcher to screen all participants for eligibility. Direct patient recruitment but this may not enable engagement of the most deprived families with whom HVs have regular contact. |
| 3. Was recruitment successful? | Recruitment target not met despite extending the recruitment period extended from 3 to 5 months. Challenges at individual and team level. | 66 parents recruited in 5.5 and 7 months. Not all HVs who were trained took part in the study (22/28). Some did not recruit parents to the study. | Funding for translation and interpreting services. Preproject discussions about the impact of professional gatekeeping with HVs. Further research is needed to identify barriers and enablers to research recruitment by HVs so that evidence-based interventions to improve recruitment can be developed. |

Continued

**Table 1** Continued

| Methodological issue | Findings | Evidence | Strategies for improvement |
|---|---|---|---|
| 4. Did eligible participants consent? | Low conversion to consent. | 226 infants were eligible. 88 (38%) parents refused to participate. 138 parents were identified as being interested in the study. A failure of communication between researchers and some HVs meant that 16 parents who had given their permission to be contacted were not followed-up. Researchers were unable to get responses to their telephone calls from 22% (30/138) of parents who initially expressed interest in the study. 66 (48%) consented to participate. | Improve communication pathways between HVs and research team. Research team more visible in child health clinics to ensure they are recognisable to parents. |
| 5. Were participants successfully randomised and did randomisation yield equality in groups? | Not assessed | | |
| 6. Were blinding procedures adequate? | Not assessed | | |
| 7. Did participants adhere to the intervention? | Not all parents had a ProAsk assessment/viewed the therapeutic wheel. | 56/66 infants received the risk assessment. Interviews with HVs revealed that all 10 parents who did not receive an assessment were not at home when they called. 21/53 were found to be at above population risk but only five of these received the therapeutic wheel intervention per protocol. Process evaluation revealed that n=4 parents received the assessment but did not receive the risk score feedback. | Better engagement with HVs around the project aims and objectives, and provision of HV time for ongoing engagement with and support from research team about intervention delivery. Audit of study processes by research staff and feedback to HVs to increase adherence to the intervention. Increase accessibility to research team for participants with concerns/questions. |
| 8. Was the intervention acceptable to participants? | Some parents and HVs did not take part in the study. Low number of parents consenting suggest some were concerned about the intervention | 6/28 HVs declined to take part in the study. Qualitative interviews suggests they were wary of raising issue of infant weight but adjusted their practice to incorporate ProAsk. 88 (38%) parents refused to take part following initial interest. Parent participants found the intervention acceptable. | Determine in further qualitative analysis the particular issues or components of the intervention that HVs were uncomfortable with. |
| 9. Was it possible to calculate intervention costs and duration? | Not assessed | | |

Continued

**Table 1** Continued

| Methodological issue | Findings | Evidence | Strategies for improvement |
|---|---|---|---|
| 10. Were outcome assessments completed | HVs were able to complete risk score on 56 participants.<br>Follow-up questionnaire completion rate low. The number of returned questionnaires was much higher for parents of infants found to be at above population risk. | See table 2 for more details of outcome data. 34/66 parents completed follow-up questionnaire, of which 29 were usable.<br>15 (71%) of parents whose infants were at higher risk returned their follow-up questionnaire. Of those at population risk, 19 (42%) returned the follow-up questionnaire. | Audit and feedback by research staff to HVs to increase outcome assessments.<br>Recontact parents for reminders to complete outcome questionnaires. |
| 11. Were outcomes measured those that were the most appropriate outcomes? | The proposed primary outcome measure primary outcome measure of weight-for-age z-score.<br>Missing anthropometric data mainly in relation to infant length and head circumference at both baseline and follow-up.<br>Low numbers of parents recorded the date of the anthropometric measures on the questionnaires. Of those that did there was a date discrepancy between completion of the self-report aspect of the questionnaires (by parent at home) and the associated anthropometric measures (collected in clinic).<br>The proposed secondary outcomes were parenting self-efficacy, maternal feeding style, infant diet and exposure to physical activity/sedentary behaviour. | Number of infant anthropometric measures completed<br>*Baseline (total n=66)*<br>Weight n=64, length n=26, head circumference n=34.<br>*Follow-up (total n=34)*<br>Weight n=28, length n=15, head circumference n=14.<br>The anthropometric measures on the baseline questionnaire that were dated (n=13) were recorded on average 22 days earlier than the parent report data. The anthropometric measures on the follow-up questionnaire that were dated (n=8) were recorded on average 15 days earlier.<br>All follow-up questionnaires that were returned were completed.<br>Cronbach's alpha for the measures was >0.5 | Train research team in infant anthropometric measurement and include resources for a follow-up visit by researchers to obviate the need for self-report of infant anthropometrics at baseline and follow-up.<br>Ensure anthropometric measurement timing fits with HV service schedule. |
| 12. Was retention to study good? | Retention was low. | 34/66 (52%) of parents completed follow-up questionnaire. The average time to complete was 28 weeks (min 23, max 39 weeks).<br>Because the intervention was not delivered per protocol, there was a time lapse between the assessment and follow-up.<br>12 parents agreed to post-study qualitative interviews. | Strategies to maintain participant engagement in study are needed for a larger trial. For example, updates via website, newsletter or text messaging. |
| 13. Were the logistics of running a multicentre trial assessed? | One centre recruited better than the others. This team work well together and there were low levels of staff change and conflict. | Site 1 recruited n=39/109 eligible infants (36%), site 2 n=15/83 (18%), site 3 n=3/10 (30%), site 4 n=9/38 (24%). | Training to include team working around peer support, conflict, etc |

Continued

**8**

**Table 1** Continued

| Methodological issue | Findings | Evidence | Strategies for improvement |
|---|---|---|---|
| 14. Did all components of the intervention work together? | There were some difficulties blending the ProAsk risk assessment and the motivational approach. The motivational interviewing training was too early and insufficient for some HVs. | ProAsk delivered in its entirety to most parents. Only five HVs completed the goal setting stage resulting in a failure to offer follow-up care to parents of infants identified as at above average risk of overweight. Mismatch between MI training and implementation timing. | Communication training for HVs in raising risk and motivational approaches bespoke to childhood overweight prevention. |

HV, health visitor; ProAsk, Proactive Assessment of Obesity Risk during Infancy.

about their own weight and a further seven did so for altruistic reasons. Eleven parents were willing to be randomised for a future trial around identification and intervention with infants at future risk of overweight.

### Consent

Of the 138 parents who gave permission to be contacted by the researchers, 66 (48%) provided written consent (figure 3). Interviews with HVs suggested that one reason for the low conversion rate to written consent was that parents were wary of accepting a telephone call or arranging a home visit with unknown researchers.

#### HV N9

*So I think that then when I said someone else would come in after me, some families were not keen to take part. Half our battle is for us to get in, then when I said someone else, I found that was hard.*

### Adherence to intervention

In total, 56/66 infants had their overweight risk score calculated. Interviews with the HVs suggested that the main reason that 10 parents did not receive their risk assessment was that they were not at home when the HV came to deliver ProAsk.

#### HV C22

*A couple of them I've been to their house at the designated time and they haven't been there so I haven't revisited them. Because you know, if you go out and see them and they're not there what do you do? You perhaps have to chase them up but to honest I haven't had the time.*

An important element was for HVs to feedback the overweight risk score to parents, but four parents were not aware of having received this feedback or were uncertain as to what it meant for them. Four HVs interviewed reported difficulties feeding back to parents the overweight risk score.

Although goal-setting and follow-up contact was recommended for infants identified as being above population risk of overweight, this did not always take place. Goal-setting around behaviour change was recorded for only 5 of the 21 parents whose infants were at above population risk. HV interviews confirmed that of the 11 HVs who had conducted a ProAsk assessment and were interviewed, 7 had shown parents all elements of the wheel rather than focussing on one specific area. There was little evidence that MI had been used to facilitate goal setting and behaviour change. Three of the HVs interviewed had used the therapeutic wheel to provide information to all participants, irrespective of their risk score status.

#### HV C22

*I know when I did the actual wheel, if you like, you said to discuss one topic, we ended up discussing them all. Because all of those topics are covered in health visiting anyway, to me it didn't feel right that we talked about diet without exercise and feeding cues.*

**Table 2**  Descriptive data for n=53[1] participants who completed ProAsk assessment at baseline

| Demographic factors (n=53) | At population risk (<10%) | Above healthy risk (≥10%) | Total |
|---|---|---|---|
| Gender | | | |
| Boy (%) | 15 (46.9) | 12 (57.1) | 27 (50.8) |
| Girl (%) | 17 (53.1) | 9 (42.9) | 26 (49.1) |
| Child age (week) | 9.84 (1.5) | 10.35 (2.3) | 10.04 (1.9) |
| Birth weight (kg) | 3.28 (0.5)* | 3.86 (0.4)* | 3.51 (0.5) |
| Weight-for-age z-score | −0.67 (0.6)* | 0.32 (0.5)* | −0.26 (0.7) |
| Rapid weight gain (<0.67 SD) | | | |
| No (%) | 30 (93.8) | 17 (80.9) | 47 (88.7) |
| Yes (%) | 2 (6.2) | 4 (19.1) | 6 (11.3) |
| Smoking in pregnancy | | | |
| No (%) | 31 (96.9) | 21 (100) | 52 (98.1) |
| Yes (%) | 1 (3.1) | 0 (0) | 1 (1.9) |
| Maternal prepregnancy BMI (kg/m$^2$) *2 missing values* | 24.8 (7.2) | 27.7 (8.7) | 25.9 (7.9) |
| Paternal BMI (kg/m$^2$) *15 missing values* | 26.5 (4.6)* | 30.1 (4.2)* | 28.0 (4.8) |
| Feeding choice | | | |
| Exclusive breast feeding (%) | 11 (34.4) | 11 (52.4) | 22 (41.5) |
| Mixed formula and breast (%) | 4 (12.5) | 2 (9.5) | 6 (11.3) |
| Formula only (%) | 17 (53.1) | 8 (38.1) | 25 (47.2) |
| Mother marital status | | | |
| Married/living with partner (%) | 29 (93.6) | 21 (100) | 50 (96.2) |
| Single/separated (%) | 2 (6.4) | 0 (0) | 2 (3.8) |
| Mother employment status | | | |
| Unemployed (%) | 3 (9.4) | 4 (19.1) | 7 (13.2) |
| Part-time (%) | 4 (12.5) | 5 (23.8) | 9 (17.0) |
| Full-time (%) | 25 (78.1) | 12 (57.1) | 37 (69.8) |
| Education of mother | | | |
| General Certificate of Secondary Education (%) | 12 (37.5) | 8 (38.1) | 20 (37.7) |
| A levels (%) | 7 (21.9) | 2 (9.5) | 9 (17.0) |
| Degree (%) | 13 (40.6) | 9 (42.9) | 22 (41.5) |
| Unknown (%) | 0 (0.0) | 2 (9.5) | 2 (3.8) |
| Number of children | | | |
| One (%) | 11 (34.4) | 7 (33.3) | 18 (33.9) |
| Two (%) | 9 (28.1) | 6 (28.6) | 15 (28.3) |
| Three or more (%) | 5 (15.6) | 5 (23.8) | 10 (18.9) |
| Unknown (%) | 7 (21.8) | 3 (14.3) | 19 (18.9) |
| Ethnicity of child | | | |
| White British (%) | 28 (87.5) | 19 (90.5) | 47 (88.7) |
| Non-white British/mixed/other (%) | 4 (12.5) | 2 (9.5) | 6 (11.3) |
| ProAsk risk score | 5.05 (2.3)* | 19.4 (8.7)* | 10.7 (9.1) |
| Income deprivation affecting children, 2015 (n=56) | | | |
| Quintile 1 (%) | 3 (9) | 5 (24) | 8 (15) |
| Quintile 2 (%) | 4 (13) | 6 (29) | 10 (19) |
| Quintile 3 (%) | 14 (44) | 8 (38) | 22 (42) |

Continued

**Table 2** Continued

| Demographic factors (n=53) | At population risk (<10%) | Above healthy risk (≥10%) | Total |
|---|---|---|---|
| Quintile 4 (%) | 8 (25) | 2 (10) | 10 (19) |
| Quintile 5 (%) | 3 (9) | 0 (0) | 3 (6) |

N=56 infants had a ProAsk assessment but three participants had incomplete data transfer from HV to research team.
Categorical variables are numbers and proportions. Continuous variables are means and SD.
*p<0.05.
BMI, body mass index; HV, health visitor; ProAsk, Proactive Assessment of Obesity Risk during Infancy.

### Intervention acceptability

A total of 88 parents declined participation. Eight out of 12 parent participants found ProAsk acceptable and were positive about its digital functionality. One parent expressed disappointment with ProAsk. There was evidence that ProAsk helped to engage parents and avoided information overload.

### Parent C8

*I thought the information in there was really nice and visual actually, because sometimes you can hear a lot of information and it is sort of difficult to absorb it and there was quite a lot of it, it was nice to have something in front of you as well as you were having that discussion. A sort of a visual prompt you could refer back to.*

Six out of 28 HVs did not take part in the study. Eight HVs expressed initial concerns about the unintended consequences of communicating overweight risk status, particularly at infant age 3 months, which some considered too early for personalised risk communication.

### HV N5

*I mean when I first heard about the research I was quite concerned initially because I had visions of mothers sort of starving their babies that's more of a risk than over feeding a baby at that sort of very early age, delicate age when they're so young.*

However, HVs also recognised the potential benefits of early intervention to prevent overweight with seven stating they found the therapeutic wheel engaging to use and containing useful information. One HV suggested that the intervention was disappointing.

### Outcome assessment

Table 2 shows the demographic and participant characteristics at baseline stratified by overweight risk status. At the 10% risk threshold, 32 (60%) infants were at average population risk and 21 (40%) were above. There was a statistically significant difference in birth weight (3.28 vs 3.86) and weight-for-age z-score (−0.67 vs 0.32) between the infants that were at population risk and those above population risk at baseline. There was also a significant difference in paternal BMI (26.5 vs 30.1 kg/m$^2$) but not prepregnancy maternal BMI or smoking status.

### Selection of outcomes

The parent self-report measures were completed fully by 85% of respondents. However, there was missing data for infant length and head circumference because these measures are not routinely recorded in parent-held infant records. Cronbach's alpha for the parenting self-efficacy[40 41] and maternal feeding style (IFQ)[42] measures all exceeded >0.5, indicating acceptable internal consistency.[45 46]

Four parents described how completing the baseline questionnaire had prompted them to change their behaviour around infant opportunities for active play and sedentariness. This was an unexpected measurement effect.

### Parent C16

*When I filled in the questionnaire, at least the first time I filled them in, there were a few that made me think about how I could change it. For example there was a question about how much tummy time the baby gets. And I'd never really thought of that as a form of exercise, which I then started to do. And it made me more aware of trying to get my son that tummy time. It made me think it's not all about what they eat, it's about, well exercising the calories off.*

### Study retention

In total, 34/66 parents returned the follow-up questionnaire at 6 months (51% retention rate). Fifteen (71%) of parents whose infants were at higher risk returned their follow-up questionnaire. Three parents had not received the intervention. All parents (n=12) invited to participate in poststudy interviews agreed to take part.

### Logistics of multicentre trial

One site recruited more parents than the others (see table 1). HVs from this site were able to overcome the challenges that occurred in the early stages of the project through team working and reaching out to the researcher for support. The HVs from the other sites talked about the teams or their own resistance to the study because they felt their geographical area was unsuitable for the study.

### HV C43

*Our administrator was brilliant; I knew you were on the end of the phone; I had support from my peers, if we didn't know how to do something we worked it out between us.*

### HV N20

*It was just said that the area has been chosen so that was fine. I just think that it was the wrong area, absolutely totally the wrong area.*

| Table 3 Content analysis of qualitative data | No. of participants reporting this theme | No. of references to this theme |
|---|---|---|
| **(a) Health visitors (HVs) (N=15)** | | |
| Eligibility | | |
| Perception that the study is inappropriate for families where there are maternal mental health concerns or safeguarding issues | 8 | 13 |
| 6–8 weeks is a difficult time for HVs to approach parents | 8 | 14 |
| Language barriers problematic | 9 | 18 |
| Recruitment | | |
| HV participated at manager's request | 9 | 15 |
| HV workload made it difficult to prioritise study | 5 | 6 |
| Wariness about raising the study with some parents because of expectations that they might be offended or overloaded | 6 | 6 |
| HV engagement with project was supported by positive professional relationships | 4 | 11 |
| Belief that more educated parents were more interested in the study | 6 | 9 |
| Consent | | |
| Belief that some parents need a relationship with a professional before they will engage/permit home access | 6 | 11 |
| Belief that parental wariness of unknown researcher negatively impacted on the numbers of participants giving written consent | 4 | 7 |
| Adherence to intervention | | |
| Difficulties feeding back overweight risk score from programme to parents | 4 | 8 |
| Therapeutic wheel used with all parents | 3 | 5 |
| All elements of the wheel discussed | 7 | 12 |
| Intervention acceptability | | |
| Concern about unintended consequences of overweight risk identification | 8 | 14 |
| Belief that early prevention is better than later management | 9 | 13 |
| Belief that timing of ProAsk personalised risk communication for infants is too early for parents | 9 | 17 |
| ProAsk is engaging to use | 7 | 18 |
| ProAsk wheel (on tablet) contained useful information | 7 | 9 |
| ProAsk tablet a disappointment | 1 | 1 |
| 3–4 months is an appropriate time for this intervention | 3 | 3 |
| Components of protocol working together | | |
| Belief that HVs already do this work | 7 | 10 |
| Training day should have been closer to study start date | 7 | 9 |
| Lack of tablet device during training a problem | 5 | 7 |
| Initial lack of confidence in explaining study to potential participants | 4 | 6 |
| HV misunderstood study protocol | 5 | 6 |
| ProAsk on tablet a challenge for novel users | 8 | 18 |
| Belief that ProAsk risk assessment should be accompanied by intervention and input by HV | 6 | 21 |
| Team work and practice improved skills in using tablet | 6 | 15 |
| **(b) Parents (N=12)** | | |
| Recruitment | | |

**Table 3** Continued

| | No. of participants reporting this theme | No. of references to this theme |
|---|---|---|
| Parent felt study recruitment processes acceptable | 11 | 13 |
| Parent felt well informed about the study | 10 | 11 |
| Parent participated because of own weight issues/issues with family eating patterns | 7 | 12 |
| Parent participated for altruistic reasons | 7 | 12 |
| Parent willing to be randomised to participate in future trial | 11 | 11 |
| Adherence to intervention | | |
| No recall of, or uncertainty about, feedback of personalised overweight risk score for baby | 4 | 7 |
| Raised awareness/change in perception in response to overweight risk feedback and intervention | 3 | 3 |
| Parent reports no behaviour change following ProAsk | 4 | 5 |
| Intervention acceptability | | |
| Parental belief that they are already doing the right thing | 6 | 9 |
| Belief that early prevention is better than cure | 3 | 5 |
| Receiving risk score was upsetting | 1 | 1 |
| Receiving risk score was a relief | 2 | 2 |
| Parent would have liked more, eg, app, website, ongoing information | 3 | 5 |
| Outcome assessment | | |
| Questionnaire components a cue to behaviour change | 4 | 5 |
| Components of protocol working together | | |

## Components of protocol working together

The low level of fidelity suggests there were incompatibilities between the risk assessment and preventative strategies within ProAsk. HVs found it difficult to find the time for additional home visits to complete the behaviour change aspect of the intervention.

### HV C22

*I have to say we are a bit stretched for time, we're short of staff, and it is another visit that we have to fit in on top of everything else. So from that point of view, it was a bit stressful I suppose.*

## DISCUSSION

The aim of the study was to determine the feasibility and acceptability of conducting an RCT of ProAsk. Parents and HVs found the study processes acceptable and ProAsk engaging. However, HVs had reservations about assessing and communicating overweight risk to parents of young infants. Overall recruitment to the study was lower than expected. Poor conversion of potential participants to consent resulted in the study failing to meet the recruitment target. There were problems with protocol adherence and intervention fidelity, with some parents not receiving all elements of the intervention.

This study was conducted in areas identified as being socially deprived because childhood obesity is more prevalent.[2 3] While recruitment was disappointing, the IDACI[44]

scores for the infant participants show that more socio-economically deprived households were included. The fact that 40% of infants recruited were assessed as being at risk of overweight and that 71% of these returned their follow-up questionnaire demonstrates that our target group was sampled. Recruiting participants from these areas is known to be challenging.[47] Other research studies have used opt-in[48] or financial incentives[49] to improve recruitment in socially disadvantaged areas. However, the potential for parental stigmatisation[50] make such strategies less applicable to studies of overweight prevention. Language was a significant barrier to recruitment, and future research will need to ensure that interpreting and translation service are resourced.

The sample contained a relatively high proportion of participants with degree-level education. HV logs of 6–8 weeks visits and interviews with HVs indicated that some groups of parents such as those with a history of mental health concerns, were not approached about the study, even if they were eligible for participation, because of concerns about their ability to deal with study burden. There is evidence from other settings that professional gatekeepers do not approach all participants eligible for healthcare research.[51–54] To inform our future study design, we need to understand how to improve participant identification and recruitment. Therefore, we are currently conducting a study to identify the UK HVs'

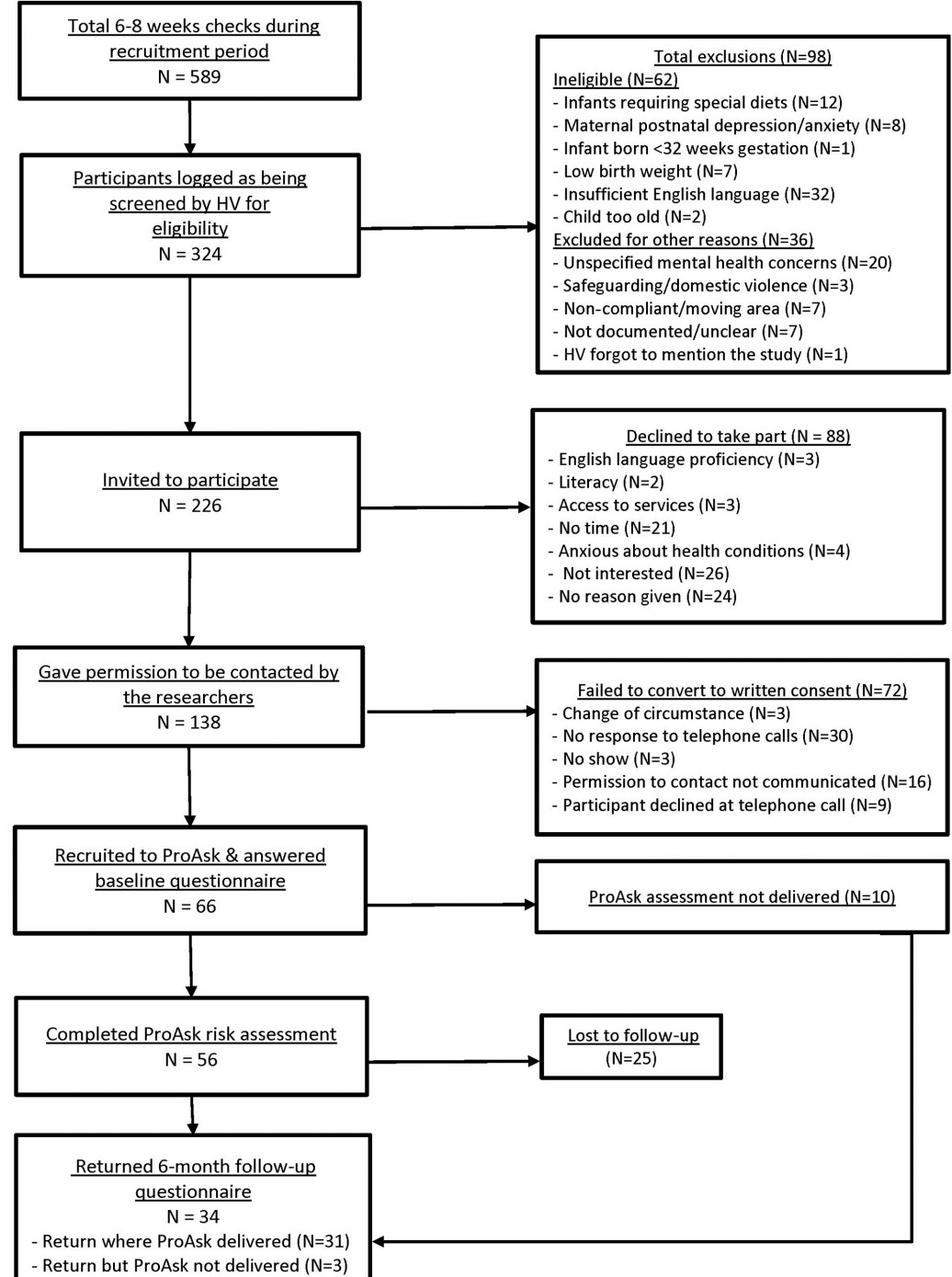

**Figure 3** Study recruitment flow chart. HV, health visitor; ProAsk, Proactive Assessment of Obesity Risk during Infancy.

perspectives on the enablers and barriers to research recruitment. It will report in late 2017.

HVs were wary about risk assessment and some had anxieties about raising the topic of weight with parents. Ten infants did not receive the risk assessment which was explained in terms of parents being unavailable but may also reflect HVs hesitancy. Problems with the technology in the field resulted in the data from three infants not transferring from tablet devices to the research team. An alternative or complementary approach could be to use routine clinical data from both parents and children for

anthropometrics. However, this could potentially reduce parental engagement and understanding of the activity of overweight risk assessment. The next phase of the study will explore whether HVs are best placed to undertake risk assessment discussions with parents and if so what training will ensure parents receive accurate information. Most HVs showed parents all the preventative information available on the therapeutic wheel rather than guiding them towards their own goals for behaviour change. HV service schedule advises them to use a motivational approach,[55] but it seemed to be challenging

for HVs to use this approach in this study. Additional bespoke training building on their existing knowledge of MI, would help HVs to deliver the behaviour change components of the intervention as intended.

Parents found ProAsk engaging and the use of digital technology acceptable. Other studies have shown similar acceptability and engagement with digital interventions resulting in improved retention of information and advice that had not been retained following a consultation.[56 57] The ProAsk therapeutic wheel could be adapted to provide accessible digital information for parents and carers which would address their request for ongoing information in digital format.

The researchers were able to collect data on outcomes of interest from parents at the times specified in the protocol. A minority of parents reported that the questionnaire items around infant activity and sedentary behaviour led them to consider behaviour change which may be a contamination risk for a future RCT. Respondent agreement between the validated outcome measures varied from poor to good, with higher levels of internal consistency in the follow-up questionnaire.

## CONCLUSIONS
The study identified significant problems with study recruitment and protocol adherence. Many of these problems could be addressed by employing dedicated researchers to screen and recruit participants. Although the intervention was acceptable to most parents and HVs interviewed, the fidelity of delivery was disappointing. There was limited evidence to support the feasibility of adding ProAsk to HV's role without significant additional resources. A future study could evaluate ProAsk as a stand-alone, parent-led digital intervention or as a HV-supported, parent-led intervention.

**Author affiliations**
[1]Professor of Public Health, Faculty of Health, Social Care & Education, Anglia Ruskin University, Cambridge, UK
[2]NIHR Research Fellow, Division of Primary Care, University of Nottingham, Nottingham, UK
[3]Research Fellow, Institute of Mental Health, University of Nottingham Innovation Park, Nottingham, UK
[4]Division of Nutritional Sciences, Associate Professor of Behavioural Nutrition, School of Biosciences, University of Nottingham, Nottingham, UK
[5]Professor of Primary and Pre-hospital Health Care, Community and Health Research Unit, School of Health and Social Care, University of Lincoln, Lincoln, UK
[6]Consultant Paediatrician, Nottingham University Hospitals Trust, Nottingham, UK
[7]Professor of e-Learning and Health Informatics, School of Health Sciences, University of Nottingham, Nottingham, UK
[8]Lead Health Visitor for Infant Nutrition, Nottingham City Care Partnership, Nottingham, UK
[9]Specialist Public Health Dietician, Nottingham City Care Partnership, Nottingham, UK
[10]Senior Lecturer in Public Health, Faculty of Medical Sciences, Anglia Ruskin University, Cambridge, UK
[11]Consultant in Public Medicine (Lead for Children), Cambridgeshire and Peterborough Public Health Directorate, Cambridge, UK
[12]Professor of Health Psychology, Institute of Mental Health, University of Nottingham Innovation Park, Nottingham, UK

**Acknowledgements** The authors are grateful to the parents and members of the health visiting teams who took part in this study. They particularly wish to thank Catherine Shiels and Fleur Seekins who managed the health visiting service in the study sites.

**Contributors** SAR, chief investigator for study and lead author, obtained NHS ethics permission, assembled team, accessed gatekeepers, project managed the study, wrote the first draft of the manuscript. JR responsible for day-to-day study coordination at two sites, recruited participants, collected and analysed data, contributed to the drafting of the manuscript. SW responsible for data processing and analysis, preparing quantitative tables and results, editing and preparation of manuscript drafts. JAS responsible for day-to-day study coordination at two sites, recruited participants, collected and analysed data and contributed to a revision of the manuscript. JAS advised on study design, was involved in data analysis and interpretation of findings and contributed to revision of the manuscript. ANS advised on study design, was involved in interpretation of findings and contributed to a revision of the manuscript. DN advised on study design, was involved in interpretation of the findings and contributed to a revision of the manuscript. HJW responsible for the design and development of the ProAsk digital tool, contributed to study design and revision of the manuscript. PA contributed to the content of the therapeutic wheel and the information that accompanied it. Contributed to the coordination of the training for health visitors (HVs) and feedback meetings. Provided local Trust data. VW contributed to the content of the therapeutic wheel and the information that accompanied it and to revision of the manuscript. FMc delivered HV training in motivational interviewing, contributed to text within ProAsk digital tool and revision of the manuscript. RL contributed to study design and revision of the manuscript. CG, local principal investigator, contribution to study design, development of ProAsk digital tool, study management and revision of the manuscript.

**Funding** This work was supported by the Medical Research Council – Public Health Intervention Development Scheme, grant number PHIND 01/14-15.

**Competing interests** None declared.

**Ethics approval** East of England NHS REC.

**Provenance and peer review** Not commissioned; externally peer reviewed.

**Data sharing statement** There is unpublished data in relation to the parents responses to the pre-questionnaire and post-questionnaire, which can be obtained by contacting the chief investigator. The unpublished qualitative data are being analysed by the authors for a separate publication.

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
