## [Reviewer comments · BMJ Open]

ARTICLE DETAILS

TITLE (PROVISIONAL)	DIGITAL TECHNOLOGY TO FACILITATE PROACTIVE ASSESSMENT OF OBESITY RISK DURING INFANCY (PROASK): A FEASIBILITY STUDY
AUTHORS	Redsell, Sarah; Rose, Jennie; Weng, Stephen; Ablewhite, Joanne; Swift, Judy; Siriwardena, Aloysius; Nathan, Dilip; Wharrad, Heather; Atkinson, Pippa; Watson, Vikki; McMaster, Fiona; Lakshman, Rajalakshmi; Glazebrook, Cristine

VERSION 1 - REVIEW

REVIEWER	Jennifer Woo Baidal, MD, MPH Columbia University Medical Center, New York, NY, USA
REVIEW RETURNED	29-May-2017

GENERAL COMMENTS	Redsell et al performed a feasibility study to inform the implementation of a future RCT to target families with infants age 6-8 weeks who are at-risk for overweight. This research is innovative as it is one of the few interventions that targets infancy for childhood overweight prevention. Furthermore, the authors targeted low-income families, who are disproportionately burdened by overweight/obesity. Novel interventions are needed during this period in the life course, which shows promise for obesity prevention but has been the target of very few interventions. The article was well-written and I enjoyed reading it. Authors described very low recruitment, which happens in real-world settings, and could serve as an important lesson for research in this field. The authors sought to provide information on the barriers to intervention recruitment and fidelity through qualitative interviews. Major comments: This manuscript does a very good job of clearly laying out limitations to recruitment and intervention implementation. However, I think this article could be greatly strengthened by including a "blueprint" to advance the field of early life obesity prevention research and should include a summary of lessons learned that could be used to improve recruitment in other studies. This could be achieved by adding a paragraph or two in the discussion about how each major recruitment and implementation obstacle could be addressed in future studies, and a table that summarizes how the methodological issues encountered by the team. - The limitations section could be further strengthened by including a discussion about why self-report of parent and infant anthropometrics is necessary and acceptable in this particular setting. (i.e., Why can't the health visitors use data from clinic visits?)
--

	Minor comments:  - Page 11, line 28: Please clarify the term "teething problems". - Table 2 and related text: Please clarify whether parental BMI is at the time of assessment or based on pre-pregnancy BMI.
--	--

REVIEWER	Senthil K Vasan Oxford Centre for Diabetes, Endocrinology and Metabolism (OCDEM), University of Oxford, Churchill Hospital, Oxford. UK OX3 7LE
REVIEW RETURNED	26-Jun-2017

GENERAL COMMENTS	The authors present the results of a feasibility study of an interactive, educational programme delivered via mobile technology for proactive obesity risk assessment among parents and health visitors. The study highlightst the public health importance of technology based obesity risk assessment system in community. The paper is well-written and presents a case for a future study incorporating the current design on a larger scale, identifies the challenges and propose solutions to be undertaken in future setting I have few comments to add – While one of the outcome measures as stated is “physical activity exposure”, I do not find any results relating to this. Was this not done in the feasibility study? If not why? In tables 3a and 3b, the authors should provide foot note as to what they refer as “no of references” Are there any specific approaches that will be undertaken to improve HV’s training strategy in future? What strategies will be taken to improve recruitment?
---

VERSION 1 – AUTHOR RESPONSE

BMJ Open - Decision on Manuscript ID bmjopen-2017-017694

Reviewer 1
Jennifer Woo Baidal, MD, MPH

Redsell et al performed a feasibility study to inform the implementation of a future RCT to target families with infants age 6-8 weeks who are at-risk for overweight. This research is innovative as it is one of the few interventions that targets infancy for childhood overweight prevention. Furthermore, the authors targeted low-income families, who are disproportionately burdened by overweight/obesity. Novel interventions are needed during this period in the life course, which shows promise for obesity prevention but has been the target of very few interventions. The article was well-written and I enjoyed reading it.

Authors described very low recruitment, which happens in real-world settings, and could serve as an important lesson for research in this field. The authors sought to provide information on the barriers to intervention recruitment and fidelity through qualitative interviews.

Major comments:
This manuscript does a very good job of clearly laying out limitations to recruitment and intervention implementation. However, I think this article could be greatly strengthened by including a "blueprint" to advance the field of early life obesity prevention research and should include a summary of lessons learned that could be used to improve recruitment in other studies. This could be achieved by adding a paragraph or two in the discussion about how each major recruitment and implementation obstacle

could be addressed in future studies, and a table that summarizes how the methodological issues encountered by the team.

Thanks for this suggestion. We have rephrased the discussion and highlighted a few strategies for how to improve recruitment in this type of study. Moreover, we have now added a column onto Table 1 (Summary of Methodological Issues) entitled strategies for improvement, and inserted text as appropriate.

We did not set out to specifically identify ways in which recruitment could be improved in studies on early life obesity prevention research. However, we are now undertaking a survey of UK health care professionals looking at barriers and enablers to recruitment and hope to publish the findings early next year. We have indicated that this research is being conducted in the text and Table 1.

- The limitations section could be further strengthened by including a discussion about why self-report of parent and infant anthropometrics is necessary and acceptable in this particular setting. (i.e., Why can't the health visitors use data from clinic visits?)

We acknowledge that health visitors could use data from clinics visits as an alternative or complementary approach as this data would perhaps be more valid and accurate. Unfortunately this data is not easily accessible to HVs (as they are in a different clinical system and not linked to infant data). We collected and recorded data face to face with parents to facilitate engagement with this activity and understanding of risk calculations, which can be seen as difficult to understand. We found that the aspect of face to face engagement with parents was well received. User engagement with digital technology as a theme has been explored in a further paper on the communication process of ProAsk. We have added some information about this to the main text.

Minor comments:

- Page 11, line 28: Please clarify the term "teething problems".

Teething problems refer to short-term problems that occur in the early stages of a new project. This is a common phrase in the UK but to ensure understanding extends to an international audience we have changed it.

- Table 2 and related text: Please clarify whether parental BMI is at the time of assessment or based on pre-pregnancy BMI.

Pre-pregnancy weight which has been added to the text and Table 2.

Reviewer 2 **Senthil K Vasan**

The authors present the results of a feasibility study of an interactive, educational programme delivered via mobile technology for proactive obesity risk assessment among parents and health visitors. The study highlights the public health importance of technology based obesity risk assessment system in community. The paper is well-written and presents a case for a future study incorporating the current design on a larger scale, identifies the challenges and propose solutions to be undertaken in future setting I have few comments to add – While one of the outcome measures as stated is “physical activity exposure”, I do not find any results relating to this. Was this not done in the feasibility study? If not why?

We designed this study in line with the NIHR recommendations for a feasibility study https://www.nihr.ac.uk/funding-and-support/documents/funding-for-research-studies/research-programmes/RfPB/FAQs/Feasibility_and_pilot_studies.pdf. The purpose was to explore whether our proposed outcome measures were appropriate and whether this data could be collected in UK community care. The study design did not include examination of pre/post within-subjects outcomes and was not powered for us to do this. To make this clearer we have detailed the proposed primary and secondary outcome measures in the text and on Table 1.

In tables 3a and 3b, the authors should provide foot note as to what they refer as “no of references”

We have changed the column headers to make this clearer

Are there any specific approaches that will be undertaken to improve HV's training strategy in future?
We didn't explore this for this study but we have re-written the paragraph in the discussion about training to make the gaps and possible solutions more explicit.

What strategies will be taken to improve recruitment?

Thanks for this suggestion. We have rephrased the discussion and highlighted a few strategies for how to improve recruitment in this type of study. Moreover, we have now added a column onto Table 1 (Summary of Methodological Issues) entitled strategies for improvement, and inserted text as appropriate.

We did not set out to specifically identify ways in which recruitment could be improved in studies on early life obesity prevention research. However, we are now undertaking a survey of UK health care professionals looking at barriers and enablers to recruitment and hope to publish the findings early next year. We have indicated that this research is being conducted in the text and Table 1.

VERSION 2 – REVIEW

REVIEWER	Jennifer Woo Baidal Columbia University USA
REVIEW RETURNED	23-Jul-2017

GENERAL COMMENTS	The reviewers comments have been fully addressed. The new information provided strengthens the importance of this feasibility study and it will be an important addition to the literature in order to advance improvements during infancy to prevent obesity through public health systems.
--

REVIEWER	Senthil K Vasan MD, PhD OCDEM, Radcliffe Dept of Medicine University of Oxford Churchill Hospital, Oxford OX3 7LE, UK
REVIEW RETURNED	14-Jul-2017

GENERAL COMMENTS	The authors have responded to the reviewers comments satisfactorily and have made relevant changes in the manuscript. I have no further comments to add. The manuscript can be accepted in its current format.
--